# Laboratory diagnostic, epidemiological, and clinical characteristics of human leptospirosis in Okinawa Prefecture, Japan, 2003–2020

Tetsuya Kakita[1]*, Sho Okano[1¤a], Hisako Kyan[1], Masato Miyahira[1¤b], Katsuya Taira[1¤a], Emi Kitashoji[2], Nobuo Koizumi[3]*

1 Department of Biological Sciences, Okinawa Prefectural Institute of Health and Environment, Uruma, Okinawa, Japan, 2 Department of Clinical Medicine, Institute of Tropical Medicine, Nagasaki University, Nagasaki, Nagasaki, Japan, 3 Department of Bacteriology I, National Institute of Infectious Disease, Shinjuku, Tokyo, Japan

¤a Current address: Okinawa Prefectural Government, Naha, Okinawa, Japan
¤b Current address: Sanitary Affairs Division, Yaeyama Regional Public Health Center, Ishigaki, Okinawa, Japan
* kakitatt@pref.okinawa.lg.jp (TK); nkoizumi@niid.go.jp (NK)

**Data Availability Statement:** All flaB sequences are available from the DDBJ database (accession

## Abstract

### Background

Leptospirosis is considered an endemic disease among agricultural workers in Okinawa Prefecture, which is the southernmost part of Japan and has a subtropical climate, but data on the current status and trend of this disease are scarce.

### Methodology/principal findings

We conducted a retrospective study of clinically suspected leptospirosis patients whose sample and information were sent to the Okinawa Prefectural Institute of Health and Environment from November 2003 to December 2020. Laboratory diagnosis was established using culture, nested polymerase chain reaction (PCR), and/or microscopic agglutination test (MAT) with blood, cerebrospinal fluid, and/or urine samples. Statistical analyses were performed to compare the epidemiological information, clinical features, and sensitivities of diagnostic methods among laboratory-confirmed cases. Serogroups and the species of *Leptospira* isolates were determined by MAT using 13 antisera and *flaB* sequencing.

A total of 531 clinically suspected patients were recruited, among whom 246 (46.3%) were laboratory confirmed to have leptospirosis. Among the confirmed cases, patients aged 20–29 years (22.4%) and male patients (85.7%) were the most common. The most common estimated sources of infection were recreation (44.5%) and labor (27.8%) in rivers. Approximately half of the isolates were of the *L. interrogans* serogroup Hebdomadis. The main clinical symptoms were fever (97.1%), myalgia (56.3%), and conjunctival hyperemia (52.2%). Headache occurred significantly more often in patients with Hebdomadis serogroup infections than those with other serogroup infections. The sensitivities of culture and PCR exceeded 65% during the first 6 days, while the sensitivity of MAT surpassed that of culture

numbers LC642635–LC642675). All relevant data are within the manuscript and its Supporting Information files.

**Funding:** This work was supported by the Japan Agency for Medical Research and Development (funding acquisition: NK; grant number: JP21fk0108139; URL: https://www.amed.go.jp). The funders had no role in study design, data collection and analysis, decision to publish, or preparation of the manuscript.

**Competing interests:** The authors have declared that no competing interests exist.

and PCR in the second week after onset. PCR using blood samples was a preferable method for the early diagnosis of leptospirosis.

## Conclusions/significance

The results of this study will support clinicians in the diagnosis and treatment of undifferentiated febrile patients in Okinawa Prefecture as well as patients returning from Okinawa Prefecture.

### Author summary

Leptospirosis is an important but largely under-recognized public health problem in the tropics. Recreational leptospirosis is a major cause of infection; people contract leptospirosis via recreational activities in rivers or lakes. Okinawa Prefecture is located in the southernmost part of Japan, has a subtropical climate, and is a major tourist destination, with an estimated 10 million visitors annually. In this study, we revealed that the trend of leptospirosis in this prefecture has changed. Leptospirosis was considered an endemic disease among agricultural workers; however, during the last 17 years, most of the confirmed cases in Okinawa Prefecture were attributable to recreation or labor in rivers in the summer in the northern part of the main island of Okinawa and the Yaeyama region, which are designated as national parks with abundant nature. Since recreation in rivers is the most popular activity among tourists in Okinawa Prefecture, leptospirosis is a health concern not only for residents of Okinawa Prefecture but also for tourists.

## Introduction

Leptospirosis is the most common zoonotic disease and is caused by infection with pathogenic spirochetes of the genus *Leptospira*, which comprises 64 species divided into 24 serogroups and more than 300 serovars [1–4]. *Leptospira* spp. colonize the proximal renal tubules of maintenance hosts, including wild animals such as rats and wild boars, livestock such as cattle and pigs, and companion animals such as dogs and are shed in their urine [2,5,6]. Humans are percutaneously or permucosally infected with *Leptospira* spp. by direct contact with the urine of maintenance hosts or by indirect contact with soil or water contaminated with infected urine [3,6,7].

Approximately one million cases of human leptospirosis occur worldwide annually, along with 58,900 deaths [8]. Seventy-three percent of the world's leptospirosis cases and deaths occur in tropical regions [8]. Human leptospirosis is characterized by a range of clinical symptoms, ranging from mild influenza-like illnesses (such as fever, myalgia, and headache) to a severe form called Weil's disease (which presents with jaundice and acute renal failure) and leptospirosis pulmonary hemorrhage syndrome (which presents as pulmonary hemorrhage without jaundice or renal failure). The mortality rates of Weil's disease and leptospirosis pulmonary hemorrhage syndrome are >10% and >50%, respectively [9]. Since most symptomatic patients present with a mild form of the disease, with nonspecific manifestations that are easily confused with those of other infectious diseases in the tropics, such as dengue fever, malaria, and scrub typhus, leptospirosis remains a significant diagnostic challenge for clinicians [6,10]. The incubation period for leptospirosis ranges from 3 days to 1 month, with an

average of 7–12 days [6,7]. Leptospirosis can be treated with antibiotics such as doxycycline and ampicillin; thus, early treatment can reduce suffering and mortality [3,6,11–13].

Definitive diagnosis of leptospirosis is based on laboratory diagnostic methods such as *Leptospira* culture, DNA detection via polymerase chain reaction (PCR), and antibody detection by microscopic agglutination test (MAT) using paired serum samples. Choosing adequate methods and clinical samples depends on the phase of the infection. *Leptospira* spp. exist in the blood of patients for approximately 10 days after disease onset (leptospiraemic phase) [14]. *Leptospira* spp. also appear in other body fluids, such as urine and cerebrospinal fluid (CSF), a few days after disease onset [14]. For the isolation and detection of *Leptospira* spp., blood is the most suitable sample in the leptospiraemic phase, while urine is the most suitable sample after approximately 1 week after disease onset (leptospiraemic phase), when anti-*Leptospira* antibodies are produced [14]. Anti-*Leptospira* antibodies can be detected by MAT in the blood approximately 5–10 days after disease onset [14].

In Japan, leptospirosis in humans has been considered a notifiable disease based on laboratory diagnosis under the Act on the Prevention of Infectious Diseases and Medical Care for Patients with Infectious Diseases (Infectious Diseases Control Law of Japan) since November 2003. In the National Epidemiological Surveillance of Infectious Disease study, 16–76 annual cases of human leptospirosis were reported in Japan between 2004 and 2020, with an average of 32 cases per year (0.02/100,000) [15]. In Okinawa Prefecture, located in the southernmost part of Japan, which has a subtropical climate, the number of leptospirosis cases ranged from four to 43 per year (average 14.4±10.6; 1.02/100000) [16], and accounted for more than half of all cases in Japan [15]. Leptospirosis was considered an endemic disease among agricultural workers in Okinawa Prefecture; thus, an in-house leptospirosis vaccine was introduced on a remote island [17,18]. Although there are some case reports on human leptospirosis [19–22], the current status and trend of this disease remain scarce in this prefecture.

The present report describes the results of a retrospective study comparing the availability of laboratory diagnostic methods for leptospirosis and also analyzing *Leptospira* isolates and epidemiological and clinical features of confirmed leptospirosis cases in Okinawa Prefecture during the period from November 2003 to December 2020.

## Methods

### Ethics statements

Ethical clearance was not required because sample collection was performed under the Infectious Diseases Control Law of Japan and was not performed specifically for the present study. The Medical Research Ethics Committee of the Okinawa Prefectural Institute of Health and Environment for the use of human subjects exempts their reviews for the characterization of *Leptospira* isolates obtained by laboratory diagnosis performed under the law and information that has already been anonymized and cannot identify individuals.

### Surveillance, case definition, and data collection

This retrospective study involved 531 patients who were clinically diagnosed with leptospirosis by physicians at clinics and hospitals in Okinawa Prefecture. According to the National Epidemiological Surveillance of Infectious Diseases under the Infectious Diseases Control Law of Japan, clinical samples and leptospirosis surveillance data were collected, and the clinical samples were sent to the Okinawa Prefectural Institute of Health and Environment through regional public health centers for laboratory confirmation. The inclusion criterion was clinically suspected leptospirosis cases from which at least one of the clinical samples was collected, while the exclusion criterion was the cases from which no clinical samples were collected.

Since at least one clinical sample was taken from all 531 suspected cases, there were no excluded cases for laboratory diagnosis. The case definition of confirmed leptospirosis is a person with clinical symptoms compatible with leptospirosis and with laboratory confirmation of acute infection (see "Laboratory diagnostics for leptospirosis"). One of the confirmed cases was imported; the patient had contracted leptospirosis in Thailand, and the case was removed from epidemiological and clinical analyses. Seven cases were not included in the sensitivity comparison of diagnostic methods because of a lack of information.

Leptospirosis surveillance data including date of onset and sample collection, patient sex and age, patient symptoms, and epidemiological data, such as estimated infection source, occupation, and estimated area of infection, were obtained from the survey form submitted by medical institutions. The duration between symptom onset and sample collection was calculated by setting the day of onset to the first day of illness. Age was used as a continuous and categorical variable with age groups: 0–9, 10–19, 20–29, 30–39, 40–49, 50–59, 60–69, 70–79, and 80–89 years. The patients' symptoms included fever, myalgia, conjunctival suffusion, arthralgia, renal dysfunction, liver dysfunction, headache, gastroenteritis, jaundice, shock, Jarisch–Herxheimer reaction, lymphadenopathy, meningitis, disturbance of consciousness, rash, upper respiratory inflammation, and lower respiratory inflammation. The estimated infection source was classified as recreation in rivers, labor in rivers, labor or recreation in freshwater other than rivers, agriculture, and direct or indirect contact with rodents. Okinawa Prefecture was divided into five regions: northern, central, and southern regions of Okinawa Main Island, the Yaeyama region (including Ishigaki Island and Iriomote Island), and the Miyako region.

## Laboratory diagnostics for leptospirosis

Culturing of *Leptospira* spp. from blood, CSF, and urine samples was performed using liquid Ellinghausen–McCullough–Johnson–Harris (EMJH) medium and/or Korthof's medium [7], and 50 μL of clinical sample were inoculated into 10 mL of liquid medium and incubated at 30°C for two months.

For *Leptospira* DNA detection, DNA was extracted from 200 μL of blood, CSF, or urine samples using the QIAamp DNA Blood Mini Kit (Qiagen, Hilden, Germany). The extracted DNA was then subjected to nested PCR targeting *flaB* for pathogenic *Leptospira* spp. [23].

To detect anti-*Leptospira* antibodies in paired serum samples, MAT was performed using 13 serovar strains of 12 serogroups, as previously described [23]. A positive MAT result was defined as a four-fold or greater increase in titers between acute and convalescent serum samples.

## Species and serogroup identification for *Leptospira* isolates

To identify the serogroups of the 123 *Leptospira* isolates, MAT was performed using 13 antisera diluted at 1:400 to 1:102400 through two-fold serial dilutions [23].

Leptospiral DNA was extracted from 43 of the 123 isolates using the QIAamp DNA Blood Mini Kit (Qiagen) and subjected to PCR targeting *flaB* using the primers L-*flaB*-F1 and L-*flaB*-R1 [23]. The nucleotide sequences of the amplicons were determined using the BigDye Terminator v3.1 Cycle Sequencing Kit (Applied Biosystems, Foster City, CA, USA). The *flaB* sequences were then deposited in a public database (DDBJ accession numbers LC642635–LC642675).

## Statistical analysis

Epidemiological and clinical features were compared between confirmed and non-confirmed cases and between Hebdomadis serogroup-infected cases and other serogroup-infected cases.

The sensitivity of each diagnostic method was calculated as the number of positives in each method / the number of confirmed cases. A reciprocal MAT titer of 320 was regarded as positive, irrespective of acute or convalescent samples, for the analysis of the sensitivity comparison of diagnostic methods. Continuous data were analyzed using Welch's t-test, Student's t-test, or Mann–Whitney U test and categorical data using $2 \times 2$ Fisher's exact test. Odds ratios (ORs) with corresponding 95% confidence intervals were calculated to evaluate the association of patients' symptoms with each group.

## Results

In the 17-year-period from November 2003 to December 2020, a total of 388 blood, 29 CSF, and 300 urine samples from 531 patients clinically suspected of leptospirosis were subjected to laboratory diagnosis (Fig 1).

Overall, 246 (46.3%) patients were positive based on any of the three diagnostic tests for leptospirosis (Table 1). The positivity rates for culture and DNA detection were 34.8% and 34.5%, respectively, while blood samples showed the highest positivity rate in both culture (34.2%) and DNA detection (29.3%). Antibody detection using paired sera showed positivity in 49.9% (177/355) of suspected cases. All culture- and/or PCR-positive cases in the acute samples were MAT-positive in their convalescent serum samples. The diagnostic agreement between culture and nested PCR was 66%; nested PCR detected leptospiral DNA in 21 culture-negative samples, while *Leptospira* spp. were isolated from 12 PCR-negative cases.

### Epidemiological features of the confirmed leptospirosis cases

Epidemiological information on the confirmed cases is shown in Table 2. The number of confirmed cases per month (the month of patient symptom onset) was highest in September (93 cases, 38.0%), followed by August (81 cases, 33.1%) and October (26 cases, 10.6%). Patient ages ranged from three to 84 years (median [interquartile range]: 31 [20–46]), and age of 20–29 years was the most common (55 cases, 22.4%). The sex ratio was unbalanced, with 210 men

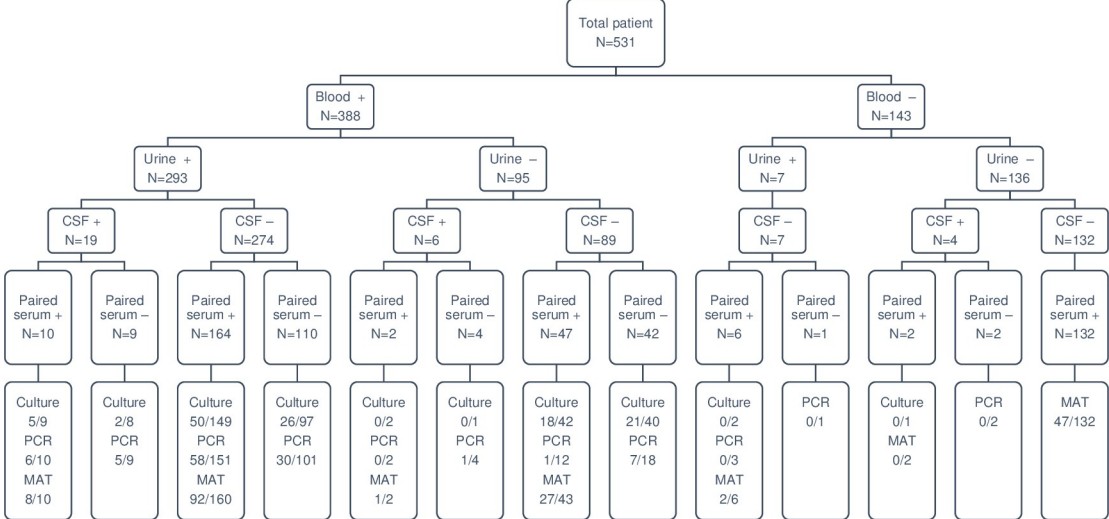

**Fig 1. Sample collection and the results for laboratory diagnostics.** The lowest boxes indicate the results of laboratory diagnostics, culture, DNA detection by *flaB*-nested PCR, and antibody detection by microscopic agglutination test (MAT). The numbers in the lowest boxes represent the number of positives / the number of patients tested. +; successful collection, -; unsuccessful collection, CSF; cerebrospinal fluid.

**Table 1. Results of laboratory diagnoses of leptospirosis among 531 patients with suspected leptospirosis in Okinawa Prefecture, 2003–2020.**

| | Diagnostics | Number of positives/number of patients tested | Positivity rate (%) |
|---|---|---|---|
| Clinical sample | | | |
| Blood | Culture | 119/348 | 34.2 |
| | DNA detection | 89/304 | 29.3 |
| Cerebrospinal fluid (CSF) | Culture | 3/21 | 14.3 |
| | DNA detection | 1/26 | 3.8 |
| Urine | Culture | 1/265 | 0.4 |
| | DNA detection | 35/271 | 12.9 |
| Paired sera | Antibody detection | 177/355 | 49.9 |
| Diagnostic method | | | |
| | Culture (blood, CSF, urine) | 122/351 | 34.8 |
| | DNA detection (blood, CSF, urine) | 108/313 | 34.5 |
| | Antibody detection (paired sera) | 177/355 | 49.9 |
| | Culture only | 18/34 | 52.9 |
| | DNA detection only | 4/26 | 15.4 |
| | Antibody detection only | 49/139 | 35.3 |
| | Culture/DNA detection | 47/116 | 40.5 |
| | Culture/antibody detection | 26/45 | 57.8 |
| | DNA/antibody detection | 4/15 | 26.7 |
| | Culture/DNA detection/antibody detection | 98/156 | 62.8 |
| Total | | 246/531 | 46.3 |

(85.7%) and 35 women (14.3%). Infections in rivers due to recreational activities such as swimming, canyoning, and canoeing (109 cases, 44.5%), and labor (68 cases, 27.8%), accounted for most of the estimated source of infection, while all the other infection sources such as agriculture, recreation or labor in freshwater other than rivers, and direct or indirect contact with rodents, were less than 10%. Most of the people who contracted leptospirosis in rivers via working were leisure guides (57 cases, 83.8%). Most were infected in the Yaeyama area (120 cases, 49.0%) and the northern part of Okinawa Main Island (96 cases, 39.2%).

## Characterization of *Leptospira* isolates and infecting *Leptospira* serogroups deduced by MAT

One hundred twenty-three *Leptospira* isolates were obtained in this study. The serogroups of all *Leptospira* isolates were identified, among which the predominant serogroups were Hebdomadis (65, 52.8%), followed by Pyrogenes (19, 15.4%) and Grippotyphosa (10, 8.1%) (Table 3).

The species of 43 *Leptospira* isolates were identified and, combined with the serological results, were classified as *L. interrogans* serogroup Hebdomadis (20, 46.5%), *L. interrogans* serogroup Autumnalis (6, 14.0%), *L. interrogans* serogroup Pyrogenes (6, 14.0%), *L. borgpetersenii* serogroup Javanica (2, 4.7%), *L. interrogans* serogroup Grippotyphosa (2, 4.7%), *L. interrogans* serogroup Icterohaemorrhagiae (2, 4.7%), *L. interrogans* serogroup Sejroe (2, 4.7%) [19], *L. interrogans* serogroup Australis (1, 2.3%), and *L. interrogans* unidentified serogroups (2, 4.7%).

The infecting serogroup was also deduced by MAT using paired serum samples, and as with the isolates, the predominant serogroup was Hebdomadis (101, 57.4%) (Table 3).

## Clinical features of the confirmed leptospirosis patients

The symptoms of patients with leptospirosis are shown in Table 4. The main clinical symptoms were fever (39.3 ± 0.9°C, 97.1%), myalgia (56.3%), conjunctival hyperemia (52.2%), arthralgia

**Table 2. Epidemiological information on cases with confirmed leptospirosis (N = 245)[a].**

| Month | Number of positives (%) |
| --- | --- |
| January | 0 (0.0) |
| February | 0 (0.0) |
| March | 1 (0.4) |
| April | 0 (0.0) |
| May | 1 (0.4) |
| June | 13 (5.3) |
| July | 18 (7.3) |
| August | 81 (33.1) |
| September | 93 (38.0) |
| October | 26 (10.6) |
| November | 9 (3.7) |
| December | 3 (1.2) |
| Age | Number of positives (%) |
| 0–9 | 13 (5.3) |
| 10–19 | 40 (16.3) |
| 20–29 | 55 (22.4) |
| 30–39 | 50 (20.4) |
| 40–49 | 37 (15.1) |
| 50–59 | 27 (11.0) |
| 60–69 | 14 (5.7) |
| 70–79 | 4 (1.6) |
| 80–89 | 5 (2.0) |
| Sex | Number of positives (%) |
| Male | 210 (85.7) |
| Female | 35 (14.3) |
| Estimated infection source | Number of positives (%) |
| Recreation in rivers | 109 (44.5) |
| Labor in rivers[b] | 68 (27.8) |
| Agricultural work | 22 (9.0) |
| Recreation or labor in freshwater other than rivers[c] | 18 (7.3) |
| Direct or indirect contact with rodents[d] | 6 (2.4) |
| Unknown | 22 (9.0) |
| Estimated area of infection | Number of positives (%) |
| Yaeyama region | 120 (49.0) |
| Northern part of Okinawa Main Island | 96 (39.2) |
| Central part of Okinawa Main Island | 9 (3.7) |
| Southern part of Okinawa Main Island | 3 (1.2) |
| Miyako region | 0 (0.0) |
| Unknown | 17 (6.9) |

[a] One of the confirmed cases was imported; the patient had contracted leptospirosis in Thailand; thus, the case was not included.

[b] Leisure guide, 57; civil engineering worker, 5; researcher, 3; fisherman, 1; unknown, 2.

[c] Civil engineering worker, 6; sanitation worker, 6; researcher, 1; unknown, 5.

[d] Sanitation worker in places inhabited by rats, 1; rat exterminator, 1; worker in a vegetable collection place inhabited by rats, 1; unknown, 3.

**Table 3. Serogroups of *Leptospira* isolates (N = 123) and infecting serogroups identified by microscopic agglutination test (MAT) in paired serum samples (N = 176) [a].**

|          | Serogroup             | Number of positives (%) |
|----------|-----------------------|--------------------------|
| Isolates | Hebdomadis            | 65 (52.8)                |
|          | Pyrogenes             | 19 (15.4)                |
|          | Grippotyphosa         | 10 (8.1)                 |
|          | Autumnalis            | 8 (6.5)                  |
|          | Australis             | 7 (5.7)                  |
|          | Icterohaemorrhagiae   | 3 (2.4)                  |
|          | Javanica              | 3 (2.4)                  |
|          | Sejroe                | 2 (1.6)                  |
|          | Unidentified[b]       | 6 (4.9)                  |
| MAT      | Hebdomadis            | 101 (57.4)               |
|          | Autumnalis            | 15 (8.5)                 |
|          | Pyrogenes             | 14 (8.0)                 |
|          | Grippotyphosa         | 9 (5.1)                  |
|          | Javanica              | 6 (3.4)                  |
|          | Australis             | 4 (2.3)                  |
|          | Ballum                | 3 (1.7)                  |
|          | Sejroe                | 2 (1.1)                  |
|          | Icterohaemorrhagiae   | 1 (0.6)                  |
|          | Multiple serogroups   | 21 (11.9)                |

[a] One of the confirmed cases diagnosed using MAT was imported; the patient had contracted leptospirosis in Thailand; thus, the result was not included.

[b] The serogroups of six isolates could not be determined due to reaction with multiple antisera.

(46.1%), and renal dysfunction (40.4%), and the occurrence of these symptoms was significantly higher in patients with confirmed leptospirosis than in those without leptospirosis. As more than half of the patients were infected with the serogroup Hebdomadis strains, the clinical symptoms of patients infected with serogroup Hebdomadis and other serogroup strains were compared. Only the occurrence of headaches was significantly higher in patients infected with the serogroup Hebdomadis. Furthermore, only one death (0.4%) was recorded in 2006 because of infection with the Australis serogroup.

## Comparison of sensitivity of laboratory diagnostic methods

The sensitivity of each diagnostic method varied depending on the duration of illness (Fig 2). During days 0–6 after onset, the sensitivities of culture and DNA detection were above 65%, while the sensitivity of antibody detection exceeded that of culture and DNA detection at one week after onset.

Compared to PCR-negative (but antibody-positive convalescent serum sample) cases, the duration between onset and sample collection in PCR-positive cases was shorter (median [interquartile range]: 5 [3–7] days vs. 3 [2–5] days, $p < 0.01$) (Fig 3). The duration between symptom onset and sample collection in blood-positive/urine-negative PCR cases was shorter than that in the blood-negative/urine-positive PCR cases (3 [2–4] days vs. 4 [3–6] days, $p < 0.01$) (Fig 4). As with PCR, the duration between symptom onset and sample collection in culture-positive cases was shorter than that in negative cases (3 [2–5] days vs. 4 [2–6] days, $p < 0.05$).

**Table 4. Clinical features of patients with leptospirosis and comparisons of clinical features in patients with leptospirosis infected with serogroup Hebdomadis and other serogroups[a].**

| Symptom | Total patients (N = 530) | | Laboratory-confirmed patients (N = 245) | | Laboratory-unconfirmed patients (N = 285) | | Odds ratio [95% CI] | Hebdomadis (N = 129) | | Others (N = 97) | | Odds ratio [95% CI] |
|---|---|---|---|---|---|---|---|---|---|---|---|---|
| | Number of positives | Positivity rate (%) | Number of positives | Positivity rate (%) | Number of positives | Positivity rate (%) | | Number of positives | Positivity rate (%) | Number of positives | Positivity rate (%) | |
| Fever | 475 | 89.6 | 238 | 97.1 | 237 | 83.2 | 6.9 [3.1–15.5] | 127 | 98.4 | 93 | 95.9 | 2.7 [0.5–15.2] |
| Myalgia | 232 | 43.8 | 138 | 56.3 | 94 | 33.0 | 2.6 [1.8–3.7] | 68 | 52.7 | 62 | 63.9 | 0.6 [0.4–1.1] |
| Conjunctival suffusion | 189 | 35.7 | 128 | 52.2 | 61 | 21.4 | 4.0 [2.8–5.9] | 66 | 51.2 | 54 | 55.7 | 0.8 [0.5–1.4] |
| Arthralgia | 183 | 34.5 | 113 | 46.1 | 70 | 24.6 | 2.6 [1.8–3.8] | 56 | 43.4 | 50 | 51.5 | 0.7 [0.4–1.2] |
| Renal dysfunction | 158 | 29.8 | 99 | 40.4 | 59 | 20.7 | 2.6 [1.8–3.8] | 44 | 34.1 | 44 | 45.4 | 0.6 [0.4–1.1] |
| Liver dysfunction | 192 | 36.2 | 88 | 35.9 | 104 | 36.5 | 1.0 [0.7–1.4] | 39 | 30.2 | 41 | 42.3 | 0.6 [0.3–1.0] |
| Headache | 136 | 25.7 | 73 | 29.8 | 63 | 22.1 | 1.5 [1.0–2.2] | 45 | 34.9 | 20 | 20.6 | 2.1 [1.1–3.8] |
| Gastroenteritis | 131 | 24.7 | 68 | 27.8 | 63 | 22.1 | 1.4 [0.9–2.0] | 34 | 26.4 | 30 | 30.9 | 0.8 [0.4–1.4] |
| Jaundice | 89 | 16.8 | 46 | 18.8 | 43 | 15.1 | 1.3 [0.8–2.1] | 22 | 17.1 | 20 | 20.6 | 0.8 [0.4–1.6] |
| Shock | 62 | 11.7 | 36 | 14.7 | 26 | 9.1 | 1.7 [1.0–2.9] | 17 | 13.2 | 16 | 16.5 | 0.8 [0.4–1.6] |
| Upper respiratory inflammation | 44 | 8.3 | 22 | 9.0 | 22 | 7.7 | 1.2 [0.6–2.2] | 11 | 8.5 | 7 | 7.2 | 1.2 [0.4–3.2] |
| Jarisch-Herxheimer reaction | 18 | 3.4 | 16 | 6.5 | 2 | 0.7 | 1.0 [0.5–1.9] | 11 | 8.5 | 5 | 5.2 | 1.7 [0.6–5.1] |
| Lymphadenopathy | 47 | 8.9 | 15 | 6.1 | 32 | 11.2 | 0.5 [0.3–1.0] | 9 | 7.0 | 5 | 5.2 | 1.4 [0.4–4.3] |
| Meningitis | 33 | 6.2 | 14 | 5.7 | 19 | 6.7 | 0.9 [0.4–1.7] | 8 | 6.2 | 5 | 5.2 | 1.2 [0.4–3.8] |
| Disturbance of consciousness | 28 | 5.3 | 13 | 5.3 | 15 | 5.3 | 1.0 [0.5–2.2] | 6 | 4.7 | 4 | 4.1 | 1.1 [0.3–4.1] |
| Rash | 48 | 9.1 | 7 | 2.9 | 41 | 14.4 | 0.2 [0.1–0.4] | 5 | 3.9 | 1 | 1.0 | 3.9 [0.4–33.7] |
| Lower respiratory inflammation | 13 | 2.5 | 4 | 1.6 | 9 | 3.2 | 0.5 [0.2–1.7] | 3 | 2.3 | 1 | 1.0 | 2.3 [0.2–22.3] |

[a] One of the confirmed cases was imported; the patient had contracted leptospirosis in Thailand; thus, the case was not included.

## Discussion

Laboratory diagnosis is indispensable, as clinical diagnosis of human leptospirosis is difficult due to a wide variety of non-specific clinical manifestations. Moreover, for accurate laboratory diagnosis, it is important to select appropriate diagnostic methods and clinical samples according to the infection phase. The findings of the current study support those of previous studies showing that PCR using blood is a preferable method for early diagnosis of leptospirosis, while the sensitivity of MAT surpassed that of PCR and culture in the second week after onset [14,24]. Previous studies have shown that the sensitivity of conventional PCR and real-time PCR using samples in the acute phase was 60%–100%, similar to that observed in the present study [24]. However, in this study, the diagnostic agreement between PCR and culture was 66%, and there were 12 PCR-negative/culture-positive cases. Samples in which PCR-negative/ culture-positive results were obtained were all blood samples. Theoretically, blood culture can detect as little as one *Leptospira* cell; thus, the number of *Leptospira* spp. may be below the limit of detection for PCR in the early acute phase. To improve the sensitivity of PCR, RNA-based (reverse transcription) PCR has been employed, which demonstrated significantly higher sensitivity than DNA-based real-time PCR using clinical samples [25,26]. In addition, *Leptospira* DNA was detected only in urine in some cases, even in the first week after onset (Fig 4), indicating that both blood and urine should be used for PCR even in the early phase of infection.

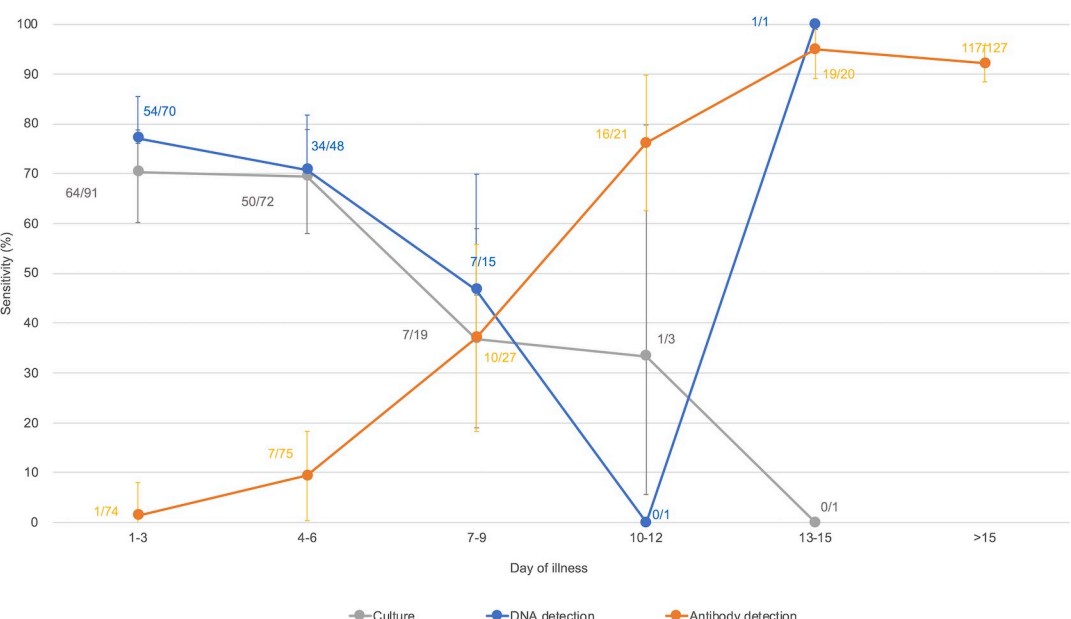

**Fig 2. Comparisons of the sensitivity of diagnostic methods on day of illness among laboratory-confirmed cases (N = 238).** Each plot indicates the number of positives / the number of samples tested. The reciprocal MAT titer 320 was regarded as positive, irrespective of acute or convalescent samples in this figure. Seven cases were not included in this analysis because of lack of information. The PCR-positive case on day 14 was a patient with diabetes. The vertical bars indicate the 95% confidence intervals.

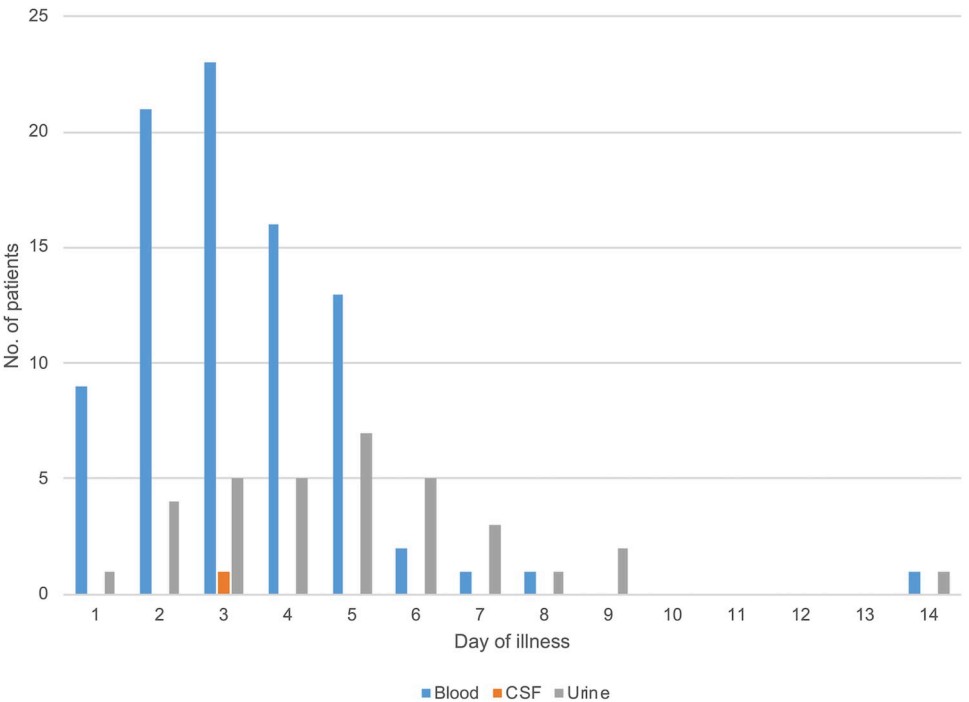

**Fig 3. Distributions of PCR-positivity among blood, cerebrospinal fluid (CSF), and urine (N = 125).** The PCR-positive results in blood and urine samples on day 14 were detected from a single patient with diabetes.

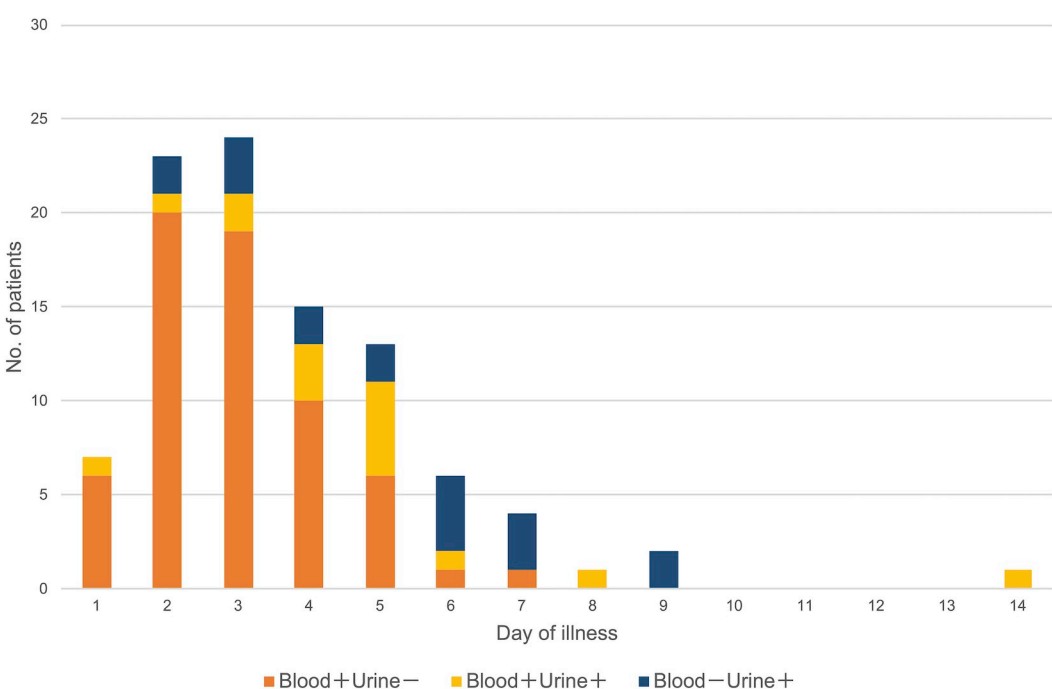

**Fig 4. Distributions of the results of *flaB*-nested PCR in blood and urine samples from single patients (N = 96).** The PCR-positive case on day 14 was a patient with diabetes.

The prevalence of *Leptospira* serogroups varies depending on the maintenance hosts, environmental conditions, and infection opportunities in specific geographic regions [11,13,27–33]. In the present study, approximately half of the causative *Leptospira* serogroups identified by culture and MAT using paired serum samples were Hebdomadis. Although serogroup Hebdomadis belongs to *L. alexanderi, L. borgpetersenii, L. interrogans, L. kirschneri, L. santarosai,* and *L. weilii* [3], all serogroup Hebdomadis isolates in Okinawa Prefecture were classified as *L. interrogans*; however, we did not identify the species of all isolates. Globally, the prevalence of serogroup Hebdomadis is not very high, accounting for 12% of serogroups detected in Japan other than Okinawa Prefecture [34], and 0–3.6% in other countries [32,33,35–37].

The clinical symptoms of patients infected with serogroup Hebdomadis were comparable with those reported previously [6,11–13,27–29,32,38–42] and did not differ significantly from those infected with other serogroups in this study except for headache, which was significantly higher in Hebdomadis-infected patients (Table 4). In previous case reports, four of four (100%) and three of four (75%) confirmed cases infected with serogroup Hebdomadis developed headache, while two out of four developed meningitis [43,44]. These findings suggest that serogroup Hebdomadis strains are more likely to cause headaches. The reported mortality rate of leptospirosis was 0–15.4% [8,11–13,27,29,30,32,33,38,39,42], compared to 0.4% in the present study due to infection with serogroup Australis. The serogroup Hebdomadis strain exhibited no lethality (low virulence) in the hamster model of leptospirosis [45,46], which may lead to a low mortality rate.

This study revealed that most leptospirosis patients were infected in rivers in the northern part of Okinawa Main Island and Yaeyama region. The sources of infection for leptospirosis vary widely depending on the country's industry, environment, and economic development, such as outdoor activities in rivers (freshwater swimming, canyoning, and fishing), contact

with rodents, livestock (animal husbandry and slaughter), stagnant water outside rivers, and agriculture [12,27,29,31–33,36,41,47]. The northern part of the main island of Okinawa and Yaeyama are rural, abundant in nature, and are designated as national parks. Outdoor activities in rivers, such as swimming, canoeing, trekking, and canyoning, are popular among both tourists and local residents. These are considered sources of infection in these areas. Although leptospirosis was not a notifiable disease before November 2003, to our knowledge, the source of leptospirosis infection in Okinawa Prefecture was 43.2% for recreation or labor in rivers and 21.0% for agricultural work from September 1988 to October 2003 (S1 Table) [48].

The number of tourists in Okinawa has been increasing and was estimated to be approximately 10 million in 2018, compared to approximately 2 million in 1988 [49]. Compared to the number before 2003, the number of patients infected in rivers increased, while the number of patients infected via agricultural work decreased. The elevation of cases was attributed to the increased numbers of tourists and their guides participating in canoeing and canyoning following the promotion of tourism in the summer months. This resulted in many confirmed cases among leisure guides, who accounted for a high proportion of the labor performed in rivers (Table 2). In addition, the proportion of males aged 20–29 years was the highest among the confirmed cases, which is attributable to the fact that the median age of the patients who contracted leptospirosis in rivers due to recreation or labor activities was 28 years, while that of patients with other infection sources was 43.5–59 years (S2 Table). The average age of patients with leptospirosis worldwide is 26.7–45 years, with individuals aged 20–29 years at high risk [8,11–13,27–33,38,40–42,47]. In addition, males comprise 61.4–92.3% of patients with leptospirosis [8,12,13,27–33,38,40,41,47]. The sex ratio of patients in the present study was comparable (83.1% to 94.4% in males) among the estimated infection sources in this study (S2 Table). These findings indicate that young people have more opportunities for infection, such as outdoor activities and occupations in rivers, but that sex imbalance was not associated with the infection sources.

In conclusion, analysis of the characteristics of human leptospirosis in Okinawa Prefecture showed that many adult males are infected with *Leptospira* spp. during recreation and labor in rivers in the summer in the northern part of the main island of Okinawa and the Yaeyama region; among them, the predominant infecting serogroup is Hebdomadis. Okinawa Prefecture is a major tourist destination, with an estimated 10 million visitors annually, including 7 million visitors from mainland Japan and 3 million from overseas. The median duration of stay for tourists is approximately 4 days [49]. Since the incubation period of leptospirosis is 7–12 days [7], most infected tourists develop the disease after they return home. In fact, leptospirosis patients who stayed and took part in water activities in a river of Okinawa Prefecture have been reported in prefectures other than Okinawa annually; however, this number is lower than the number of leisure guide patients [50,51], suggesting that many cases are overlooked in prefectures other than Okinawa and likely overseas. Since leptospirosis has nonspecific and variable manifestations, it is important for clinicians to extract the epidemiological features shown in this study and include leptospirosis in the differential diagnosis of all patients with undifferentiated febrile illness.

## Supporting information

**S1 Table. Estimated infection source of leptospirosis patients from September 1988 to October 2003 (N = 81).**
(DOCX)

**S2 Table. Comparisons of age and sex ratio of patients by estimated infection source (N = 245).**
(DOCX)

## Acknowledgments

We thank the staff of regional public health centers for collecting the clinical specimens.

## Author Contributions

**Conceptualization:** Tetsuya Kakita, Nobuo Koizumi.

**Data curation:** Tetsuya Kakita, Emi Kitashoji, Nobuo Koizumi.

**Funding acquisition:** Nobuo Koizumi.

**Investigation:** Tetsuya Kakita, Sho Okano, Hisako Kyan, Masato Miyahira, Katsuya Taira, Nobuo Koizumi.

**Methodology:** Tetsuya Kakita, Sho Okano, Hisako Kyan, Masato Miyahira, Katsuya Taira, Emi Kitashoji, Nobuo Koizumi.

**Visualization:** Tetsuya Kakita, Emi Kitashoji, Nobuo Koizumi.

**Writing – original draft:** Tetsuya Kakita, Emi Kitashoji, Nobuo Koizumi.

**Writing – review & editing:** Tetsuya Kakita, Sho Okano, Hisako Kyan, Masato Miyahira, Katsuya Taira, Emi Kitashoji, Nobuo Koizumi.

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
