## [Decision Letter · Decision Letter 0]

25 Sep 2021

Dear Dr. Tetsuya Kakita,

Thank you very much for submitting your manuscript "Laboratory diagnostic, epidemiological, and clinical characteristics of human leptospirosis in Okinawa Prefecture, Japan, 2003–2020" for consideration at PLOS Neglected Tropical Diseases. As with all papers reviewed by the journal, your manuscript was reviewed by members of the editorial board and by several independent reviewers. The reviewers appreciated the attention to an important topic. Based on the reviews, we are likely to accept this manuscript for publication, providing that you modify the manuscript according to the review recommendations. 

Sincerely,

Vasantha kumari Neela

Associate Editor

Armanda Bastos

Deputy Editor

Reviewer's Responses to Questions

**Key Review Criteria Required for Acceptance?**

**Methods**

-Are the objectives of the study clearly articulated with a clear testable hypothesis stated?

-Is the study design appropriate to address the stated objectives?

-Is the population clearly described and appropriate for the hypothesis being tested?

-Is the sample size sufficient to ensure adequate power to address the hypothesis being tested?

-Were correct statistical analysis used to support conclusions?

-Are there concerns about ethical or regulatory requirements being met?

Reviewer #1: Culture, nested polymerase chain reaction (PCR), and microscopic agglutination test (MAT) were used in this study.

Reviewer #2: At the end of the introduction it is required to specify the month of the end of the period.

The objectives of the study are clearly articulated with a clear testable hypothesis stated.

The study design is appropriated to address the stated objectives.

The sample size is sufficient to ensure adequate power to address the hypothesis being tested.

The statistical analysis used to support conclusions is correct.

There are not concerns about ethical or regulatory requirements

Reviewer #3: The study by Kakita and collaborators describe a case series of Leptospirosis in Okinawa. It is a retrospective study that describes the epidemiology of human leptospirosis in in that region during 2003-2020. This case series is an interesting compilation of clinically diagnosed Leptospirosis patients, their diagnostic laboratory results, and their epidemiological risk factors.

Although the data is present in the manuscript, the analysis would benefit from a better description of the study population, surveillance and case definitions including inclusion/exclusion criteria, and a flow chart describing the results on the tests performed (partially described in figure 1). The authors include a STROBE checklist, but this study is a case series, therefore the criteria for a cross sectional study are not applicable. I recommend rewriting the abstract accordingly, and reorganising the methods and results sections to resemble more appropriately a case series (as reference case report studies, please see Ko et al 1999 Lancet, Ciceron et al 2000 Eur J Epidemiol, Galan et al 2021 PLoS One).

**Results**

-Does the analysis presented match the analysis plan?

-Are the results clearly and completely presented?

-Are the figures (Tables, Images) of sufficient quality for clarity?

Reviewer #1: This is a straight forward epidemiologic study with clear presentation

Reviewer #2: The analysis is presented match the analysis plan.

The results are clearly and completely presented.

In Table 3. Epidemiological information on cases with confirmed leptospirosis

The column headings are offset

S1 Table. Comparisons of age and sex ratio of patients by estimated infection source.

Percentages of estimated infection source of missing male patients

Reviewer #3: The analysis is appropriate for the retrospective case report design. Figure one can include more detail about the tests performed and excluded cases.

**Conclusions**

-Are the conclusions supported by the data presented?

-Are the limitations of analysis clearly described?

-Do the authors discuss how these data can be helpful to advance our understanding of the topic under study?

-Is public health relevance addressed?

Reviewer #1: The conclusion is reasonable.

Reviewer #2: The conclusions are supported by the data presented.

Reviewer #3: The conclusions are supported, Leptospirosis should be suspected in a febrile subject and the appropriate diagnostic workup should be adjusted by days of illness to maximise the sensitivity of the available tests.

**Editorial and Data Presentation Modifications?**

Reviewer #1: a minor typo needed be fixed. 

Line 375: was to were

Line 360: are to is.

Line 361: canyoning, 

Line 384: that  those

Reviewer #2: (No Response)

Reviewer #3: (No Response)

**Summary and General Comments**

Reviewer #1: Reasonable.

Reviewer #2: The manuscript reports the laboratory, clinical, and epidemiological characteristics of human leptospirosis from 18-year-old samples in Japan. Analysis and results of laboratory samples are important to characterize the behavior of Leptospirosis in Okinawa Prefecture.

The month of the end of the study period remains to be specified.

Reviewer #3: The study is quite informative and of clinical and epidemiological value. It summarises over a decade of experience on an important endemic zoonotic disease in the described region. it also serves to monitor future dynamics of serovars, test performance, and changes in clinical presentations or risk factors for acquiring the infection. These findings can help inform local public health policies.

PLOS authors have the option to publish the peer review history of their article (what does this mean?). If published, this will include your full peer review and any attached files.

Reviewer #1: No

Reviewer #2: No

Reviewer #3: Yes: Christian A Ganoza, MD.

Figure Files:

Data Requirements:

Reproducibility:

References

---

## [Editor Report · Decision Letter 1]

12 Nov 2021

Dear Dr.Tetsuya Kakita ,

We are pleased to inform you that your manuscript 'Laboratory diagnostic, epidemiological, and clinical characteristics of human leptospirosis in Okinawa Prefecture, Japan, 2003–2020' has been provisionally accepted for publication in PLOS Neglected Tropical Diseases.

Best regards,

Vasantha kumari Neela

Associate Editor

Armanda Bastos

Deputy Editor

---

## [Editor Report · Acceptance letter]

29 Nov 2021

Dear Dr. Kakita,

We are delighted to inform you that your manuscript, "Laboratory diagnostic, epidemiological, and clinical characteristics of human leptospirosis in Okinawa Prefecture, Japan, 2003–2020," has been formally accepted for publication in PLOS Neglected Tropical Diseases.

Best regards,

Shaden Kamhawi

co-Editor-in-Chief

Paul Brindley

co-Editor-in-Chief
